# Psychometric Assessment of the Communication Skills Scale Among Peruvian Nurses and Factors Associated with Job Insecurity

**DOI:** 10.3390/healthcare12242582

**Published:** 2024-12-22

**Authors:** Gabriela Samillán-Yncio, Jhon Alex Zeladita-Huaman, Eduardo Franco-Chalco, Roberto Zegarra-Chapoñan, Iván Montes-Iturrizaga, Zulma Jeanette Rivera-Medrano

**Affiliations:** 1Academic Department of Nursing, Faculty of Medicine, Universidad Nacional Mayor de San Marcos, Lima 15001, Peru; gsamillani@unmsm.edu.pe; 2Psychology Department, Pontificia Universidad Católica del Perú, Lima 15008, Peru; eduardo.franco@pucp.edu.pe; 3Faculty of Health Science, Universidad María Auxiliadora, Lima 15408, Peru; rob.zegarra@gmail.com; 4School of Psychology, Universidad Continental, Arequipa 04002, Peru; imontes@uc.cl; 5Emergency Service, Hospital Nacional Edgardo Rebagliati Martins, Lima 15072, Peru; zulmajeanette@hotmail.com

**Keywords:** validation study, communication, nurses, job insecurity, Peru

## Abstract

Background/Objectives: The aim of this study was to evaluate the psychometric properties of the Communication Skills Scale (CSS) among Peruvian nurses, as well as the factors associated with job insecurity during the COVID-19 pandemic. Methods: We explored five models of confirmatory factor analysis for the CSS and its four subscales and assessed the convergent validity and criterion validity of the scale by analyzing its connection with job insecurity through stepwise multiple regression. We used insights from a focus group for the cultural adaptation of the scale. In the psychometric phase, 225 nurses participated through a virtual survey. Results: The psychometric analysis revealed that the CSS and its subscales have a robust internal structure—similar to the original questionnaire—and are optimally reliable in the Peruvian population. Furthermore, the results show that job insecurity was associated with contract type, communication skills, empathy, and job satisfaction. Conclusions: The CSS and its subscales are valid and reliable to be applied to Peruvian nurses. Initiatives should be undertaken to strengthen communication skills and increase job satisfaction among nursing professionals by improving their working conditions, especially in times of crisis, to reduce job insecurity and promote well-being.

## 1. Introduction

Communication allows individuals to share messages, ideas, meanings, and feelings that influence each other’s behaviors and elicit specific responses according to their social imaginaries, thus ensuring coexistence [1]. In the health context, communication is crucial because it conditions the outcome of the health–disease process [2]. Particularly, patient-centered therapeutic communication—a growing approach in health care—is essential for appropriate care quality, increased satisfaction [3], and better health outcomes [4].

Communication skills help nursing professionals establish and maintain quality therapeutic communication with patients and their families, focusing on improving their biological, emotional, and social well-being [5]. These skills are essential for building a nurse–patient therapeutic relationship, providing comprehensive care, ensuring patient safety [3,4], and enhancing job satisfaction [6].

The COVID-19 pandemic has been accompanied by an “infodemic”—an overwhelming abundance of information, including misinformation—which has significantly impacted mental health [7]. A scoping review indicated that exposure to the COVID-19-related infodemic is associated with increased levels of anxiety, depression, and stress, particularly among young adults and females [8]. This phenomenon has also resulted in heightened perceptual disorders, suicidal ideation, and negative discourse within mental health support groups [9].

Concerning psychometric instruments, some studies have validated scales to measure nurse–patient communication as a competency. A study conducted in Mexico validated a scale that identifies communication behaviors from the patients’ perspective [10], but measuring only two factors: empathy and respect. In contrast, the Interpersonal Communication Assessment Scale (ICAS) developed in the United States assesses three factors: advocacy, therapeutic use of self, and validation [11]. Since then, the ICAS has been validated in other countries, such as Spain [12]. Similarly, in Italy, the Perceived Self-Efficacy Scale in Nursing Communication was developed to evaluate beliefs regarding a nurse’s capacity to effectively manage problematic situations during communication with patients [13].

Most of the nurse–patient communication scales have been based on patient-centered humanistic approaches [14]. Along these lines, the Communication Skills Scale (CSS) was designed in Spain for nursing professionals to self-assess their communication and social skills when they interact with individuals who need health care. Thus, the CSS contributes to achieving specific diagnoses of reality and identifying opportunities for improvement [4]. Therefore, it has been used in studies that have identified the relationship between communication skills with the burnout syndrome [15], job satisfaction [16], self-efficacy [4], and job insecurity [17,18].

Furthermore, job insecurity—defined as the perceived fear of unemployment—is considered a stressor agent of a worrying job situation [19]. During the COVID-19 pandemic, a study conducted in Peru validated a scale that measures this construct as the degree of concern related to the possibility of not being able to keep a job in the future [20]. The factors that impact the perception of this fear include social aspects of work, interpersonal relationships within organizations, a connection to meaningful work, indirect issues with family [21], musculoskeletal disorders [22], perception of health risks, and adverse economic implications [23]. This construct has gained relevance because moderate levels of job insecurity can motivate nurses to adopt constructive behaviors, resulting in successful coping [24]. However, job insecurity can unfavorably impact the efficiency and efficacy of the organization [19], health, and well-being [23]. In this regard, mental health issues, such as stress, anxiety, and depression, aggravated over the course of the pandemic in Peru [25].

Likewise, communication, as a valuable personal resource, can reduce job insecurity among employees [17], flight attendants [26], and nurses [18]. This relationship can be explained through the Conservation of Resources theory proposed by Hobfoll [27]. Firstly, nurses with adequate communication skills can establish better relationships with patients, families, physicians, and colleagues (social resources), strengthening their position within the healthcare team and enhancing their job performance. Secondly, these skills allow them to effectively communicate their needs, negotiate working conditions, resolve conflicts, and demonstrate their professional value, which contributes to protecting their employment (condition resource). Thirdly, nurses with strong communication abilities can more easily access important information regarding organizational changes, development opportunities, and administrative decisions (energy resources), allowing them to anticipate and adapt more effectively to changes in their work environment.

Currently, there is a critical gap in the assessment of nursing communication skills in Peru, as no properly validated communication skills scales exist for this context. While the CSS has been applied to Peruvian nurses [28], it was utilized without appropriate cultural adaptation. This study addresses two key questions: Is the CSS a valid and reliable instrument for assessing communication skills among Peruvian nurses, and how do factors related to job insecurity influence these communication skills? This research focus is particularly relevant given that recent studies in Peru have found that one-third of healthcare workers reported job insecurity during the COVID-19 pandemic [25]. We believe that validating the CSS, which objectively assesses patient-centered communication skills, will help identify areas for improving nursing communication competencies and ultimately enhance the quality of patient care. This study aims to evaluate the psychometric properties of the CSS among Peruvian nurses, as well as the factors associated with job insecurity during the COVID-19 pandemic.

## 2. Materials and Methods

### 2.1. Study Design

This study was conducted as part of psychometric research on the CSS, starting with a process of cultural adaptation to the Peruvian context, followed by determining psychometric properties through confirmatory psychometric analysis. Subsequently, to establish its criterion validity, a multiple regression analysis was performed to identify factors associated with job insecurity.

### 2.2. Tool Development Process

This psychometric study consisted of three phases. The first one was the cultural adaptation of the CSS to the Peruvian context. The second one consisted of determining the psychometric properties of the CSS and its subscales by generating five models of confirmatory factor analysis (CFA). The third one entailed the verification of the criterion validity of the scale by exploring its connection with job insecurity during the COVID-19 pandemic.

#### 2.2.1. Phase 1: Cultural Adaptation of the CSS

The CSS is a Likert scale with response options ranging from 1 (rarely) to 6 (very frequently). It consists of 18 items divided into four subscales: empathy, communication, respect, and social skills. Since we did not find studies exploring the psychometric properties of the CSS with Peruvian nurses, we first submitted the scale to a cultural adaptation process through a virtual focus group. In that way, we examined the cultural relevance of the items, identified equivocal or culturally unsuitable terms, obtained feedback on the clarity and understandability of the questions, and discussed possible alternatives or modifications to adapt the scale to the Peruvian context [29]. The focus group was made up of ten nurses from different regions of Peru who were studying their last subspecialty year in a Peruvian university (intensive care, emergencies and disasters, public health, nephrology, and others).

The focus-group interview was conducted via Zoom version 6.0 because, in 2021, there were still restrictions in Peru on in-person meetings due to the COVID-19 pandemic. The session was carried out according to the focus-group guide prepared by the research team. The moderator (main researcher of this study) and the facilitator (nurse experienced in conducting focus-group interviews) interacted with the participants for one hour and a half. At the beginning of the session, the participants provided verbal informed consent and authorization to record. Then, the researchers explained the purpose of the meeting and the theoretical model supporting the scale and its dimensions. Subsequently, discussion of each item of the CSS was encouraged, focusing on clarity, relevance, and cultural appropriateness. The participants were also asked to explain what they understood by the terms considered in the questions and to identify words or expressions that might be problematic or confusing in the Peruvian context. Finally, participants suggested improving the wording or content of the items and including two items in the respect subscale because, unlike the other subscales, it only had three items. Therefore, we reworded twelve items (see Appendix A) and added one item to the respect subscale [19. Escucho a mis pacientes sin prejuicios, independientemente de mis creencias (19. I listen to my patients without prejudices, regardless of my beliefs)] and one item to the social skills subscale [20. Al interactuar con los pacientes/familiares en situación de crisis, busco regular emociones y resolver conflictos (20. When interacting with patients/relatives in crisis, I try to regulate emotions and resolve conflicts)].

#### 2.2.2. Phase 2: Confirmatory Factor Analysis of the CSS

##### Study Participants

We expected to obtain a root mean square error of approximation (RMSEA) for the CFA of less than 0.08, with the probability of making a type I error equal to 0.05 and a statistical power of at least 80%. Therefore, the minimum necessary sample size was 113 participants. However, considering our non-probability sampling strategy, we gathered 225 participants.

We included nursing professionals working in primary healthcare facilities or hospitals in Peru and having access to electronic devices. We excluded nurses residing in other countries.

##### Measurement Tools

The Communication Skills Scale (CSS)

The original CSS was developed and validated for Spanish nurses by Leal-Costa et al. [4]. The model demonstrates adequate goodness-of-fit indices (χ^2^(128) = 213.17; *p* < 0.001; χ^2^/df = 1.665; RMSEA = 0.053 (90% CI: 0.040–0.065); Standardized Root Mean Squared Residual (SRMR) = 0.048; Tucker–Lewis index (TLI) = 0.938; comparative fit index (CFI) = 0.949; and incremental fit index = 0.949). Furthermore, the scale shows excellent reliability (α = 0.88) and has been externally validated with self-efficacy scales.

##### Data Analysis

We initially carried out a series of exploratory and descriptive analyses for each of the variables in the sample. Subsequently, since the original scale had an adequate theoretical construct, we determined whether the factor structure of the CSS had an adequate fit. To that end, we estimated five CFA models, of which four were for each of the communication subscales and one was for the whole scale. Given the ordinal nature of the items of the instrument, we used the weighted least squares mean and variance (WLSMV) estimator. A model is considered to have a good fit when it has a non-significant chi-square or when the CFI and TLI (relative fit indicators) are greater than 0.95. Likewise, the RMSEA must be less than 0.05, and the SRMR must be less than 0.08. To determine the reliability of the scale and subscales, once their factor models were estimated, we calculated the McDonald’s omega index for each model.

#### 2.2.3. Phase 3: Determination of Factors Associated with Job Insecurity

##### Measurement Instruments and Procedures

The Perceived Job Insecurity Scale (LABOR-PE-COVID-19)

This unifactorial scale comprises four items with response options ranging from 1 = strongly disagree to 4 = strongly agree. The reliability analysis performed with the data collected in this study indicated a McDonald’s omega value of 0.88, a high level of reliability. This scale was validated for the Peruvian population by Mamani et al. in 2020 [20]. The content validity was confirmed through expert judgment (all items reported an Aiken’s V coefficient for EFA > 0.70). Additionally, its goodness-of-fit indices were appropriate for the Exploratory Factor Analysis (EFA) model (Kaiser–Meyer–Olkin = 0.780 and Bartlett’s test (654.24; df = 6; *p* < 0.001)).

In addition to communication and its dimensions, the possible factors associated with job insecurity included gender, age, place of work, contract type, remote work, history of COVID-19 infection, and job satisfaction. This last factor was assessed with the following question: “En general, ¿cómo calificarías tu nivel de satisfacción con tu trabajo o centro laboral?” (“Overall, how would you rate your level of satisfaction with your job or place of work?”).

For the psychometric phase of the study—carried out between March 2021 and September 2022—we collected data by means of a Google Forms survey. We distributed the survey through WhatsApp groups made up of nurses working in healthcare facilities and graduate nursing students enrolled in public universities.

##### Data Analysis

Finally, to determine the criterion validity of the instrument, we estimated multiple regression models to predict job insecurity. For such estimation, we employed a stepwise selection algorithm, which allowed us to determine the model with the highest predictive capacity from the input variables. For the first model, we included the following variables: gender, marital status, place of work, contract type, job satisfaction, remote work, history of COVID-19 infection, COVID-19 patient care, age, and overall communication. For the second model, we replaced overall communication with the four subscales to avoid multicollinearity problems in the model estimation. We checked the assumptions of all models for normality of residuals, homogeneity of variances, multicollinearity, and outliers. We carried out all statistical analyses in R v4.2.1 [30].

## 3. Results

A total of 225 nurses completed the survey, of which 93.3% were women. The mean age was 37 years (SD = 11.25). Overall, 58.8% of the participants reported being single, while 36.1% were married, 4.3% were separated, and 0.8% were widowed. Regarding professional characteristics, the respondents reported a mean time of work experience of 10.29 years (SD = 11.01). As for place of work, 56.9% of the nurses reported working in a hospital, 23.1% in a healthcare center, 7.1% in a polyclinic, and 12.9% in another type of healthcare facility. According to the contract type reported by the participants, 53.3% had a temporary administrative service contract; 25.5% had a tenured-employee contract; 7.5% had a permanent contract; 6.3% had a fixed-term contract; and 7.4% had a fee-for-service contract. Most of the respondents (85.1%) were working in person when the assessment was taken. In addition, 61.9% felt satisfied with their job, 32.2% felt neither satisfied nor dissatisfied, and 5.9% felt dissatisfied. Finally, 50.9% of the nurses reported that they did not have a history of COVID-19 infection.

### Description of the Psychometric Properties of the Communication Skills Scale

Table 1 shows the descriptive statistics for each of the items of the CSS. Particularly, the highest mean was for item 18 (M = 4.75, SD = 1.10), while the lowest mean was for item 16 (M = 3.87, SD = 1.09).

Given the ordinal nature of the items, during the estimation, using WLSMV adjusted, we determined that items 16 and 18 did not exhibit significant factor loadings in their respective scales or the overall factor model, with *p*-values exceeding 0.05. Consequently, we decided to remove these items from the analysis. Additionally, item 10 demonstrated a low factor loading (0.29) but was statistically significant, allowing us to retain it in the final model. These adjustments resulted in the CSS comprising 18 items (see Appendix A). Figure 1 shows the four confirmatory models estimated for each subscale. We found that all the models achieved an adequate level of fit. For the empathy subscale model, we obtained a non-significant chi-square index (X^2^ = 6.82, df = 5, and *p* = 0.24), which indicated a good fit for the model. We observed a similar scenario with the confirmatory model for the communication subscale, which also showed a non-significant chi-square index (X2 = 4.09, df = 9, and *p* = 0.91). The model for the respect subscale also obtained a non-significant chi-square index (X2 = 5.04, df = 2, and *p* = 0.08). Lastly, in the confirmatory model for social skills, we identified a significant chi-square index (X2 = 104.55, df = 3, and *p* < 0.001), suggesting that this model does not have a good fit in absolute terms. However, the relative fit indicators of this model revealed that its level of fit was adequate (CFI = 1, TLI = 1, RMSEA = 0.00, and SRMR = 0.00).

All the factor loadings of the models, as shown in Figure 1, were statistically significant (*p* < 0.001). Notably, all the loadings for empathy were greater than 0.62, confirming that the subscale is highly reliable. A subsequent analysis estimated the McDonald’s omega index for this subscale, resulting in a value of 0.82 and confirming also the high reliability of the subscale. Similarly, all the factor loadings for the communication subscale were above 0.62, obtaining a McDonald’s omega index of 0.84. These two values indicate a high reliability of the subscale. Furthermore, it was observed that item 3 of the respect subscale had a factor load of 0.55, which was the lowest in the model, and the McDonald’s omega index of this subscale was 0.77. These indicators suggest an adequate fit of the subscale. Finally, the social skills model showed that item 10 had a value of 0.29, which is considered low, although still significant. The McDonald’s omega index for this subscale was 0.60, indicating an acceptable level of reliability. In summary, the empathy and communication subscales demonstrated the highest reliability, while the social skills subscale showed the lowest.

Figure 2 shows the overall confirmatory model for the CSS. The scale has a good absolute fit according to the chi-square index (X2 = 84.44, df = 135, and *p* = 1.00), indicating a correct factor structure. All the factor loadings of the model were statistically significant (*p* < 0.001), and most of them had scores above 0.60. Only items 3 and 10 had factor loadings of 0.53 and 0.29, respectively, which are low but still significant. Finally, the McDonald’s omega index for the scale was 0.94, indicating excellent reliability.

Table 2 shows the first stepwise regression model that we used to predict job insecurity scores from the participants’ sociodemographic variables and the entire CSS. For this model, the variables included in the selection algorithm were contract type, overall communication, and job satisfaction. Regarding the contract type, the participants with a temporary administrative service contract had a higher perception of insecurity than those with a tenured-employee contract (b = 0.84, *p* < 0.001). This difference is considered to be strong according to the standardized regression coefficient (B = 0.53). In addition, the participants with a fixed-term contract also reported higher scores in job insecurity than those with a tenured-employee contract (b = 0.56, *p* < 0.01). Nevertheless, this difference is mild according to the standardized regression coefficient (B = 0.17).

Lastly, the participants with a fee-for-service contract perceived greater job insecurity than those with a tenured-employee contract (b = 0.84, *p* < 0.001). This difference is moderate according to the standardized regression coefficient (B = 0.28). Concerning the levels of communication, we found that higher scores on the CSS were connected with lower scores on job insecurity (b = −0.01, *p* < 0.01). This effect is mild according to the standardized regression coefficient (B = −0.17). As for job satisfaction, the participants who felt dissatisfied also reported higher levels of job insecurity than those who felt satisfied (b = 0.40, *p* < 0.05). This difference is mild according to the standardized regression coefficient (B = 0.12). Lastly, when observing the overall fit of the model, we found that all the variables included accounted for 25% of the total variability of job insecurity (adjusted R^2^ = 0.25, *p* < 0.01).

Table 3 shows the stepwise regression model that we employed to predict the participants’ levels of job insecurity. For this model, we took the sociodemographic variables of the participants as predictors and used the subscales of the CSS separately to identify which subscales were most related to job insecurity. The stepwise selection algorithm identified that the variables that significantly contributed to explaining the variance in job insecurity were contract type, the empathy subscale, and job satisfaction. Regarding the contract type, the results were very similar to those presented in Table 2, that is, the participants with temporary administrative service contracts, fixed-term contracts, and fee-for-service contracts had higher levels of job insecurity than those with tenured-employee contracts.

Moreover, the algorithm only chose the empathy subscale for the present model, showing that higher scores in empathy were associated with lower levels of job insecurity (b = −0.04, *p* < 0.01). This effect is considered mild according to the standardized regression coefficient (B = −0.18). No other subscales were chosen by the algorithm. Similarly, the participants with higher levels of job dissatisfaction scored higher on the job insecurity scale than satisfied participants. This model had a 26% explained variance in the response variable (adjusted R^2^ = 0.26, *p* < 0.001).

## 4. Discussion

Based on a CFA, this study reports that the CSS applied to a sample of Peruvian nurses has a similar internal structure to the original questionnaire developed in Spain [4]. Furthermore, this study shows that the CSS has excellent reliability (ω = 0.094), proving that the psychometric properties of the scale are suitable to the Peruvian context. In addition, the scale has criterion validity because its association with job dissatisfaction was confirmed. Similarly, an adequate fit was found for each of the confirmatory models estimated for the different subscales of this construct (communication skills), revealing that the subscales can be applied independently if necessary. It is worth noting that the social skills subscale showed an acceptable level of reliability (ω = 0.60), but lower than that found in the other subscales. This finding is in line with another psychometric study that validated the CSS for health professionals [31]. The reason behind this result could be the small number of items on this subscale compared to the other subscales.

The cultural adaptation of the CSS for its application to Peruvian nurses proved to be a crucial step in our study. This process made it possible to pinpoint and address subtle but significant linguistic and cultural differences, thereby improving the content validity and conceptual equivalence of the instrument in Peru [32]. Conducting the focus-group interview according to the recommendations by Squires et al. [33] entailed an in-depth exploration of the relevance and understandability of the items. Moreover, the perspective of the experienced professionals in the focus group was crucial considering that they were pursuing their subspecialty studies. This participatory approach not only enriched the instrument with specific cultural perspectives but also potentially increased its acceptability among the target population, which is essential for its effective implementation in clinical practice [34]. This process enhances both the content validity and conceptual equivalence of the scale, while also increasing its acceptance among the target population.

Regarding the analysis of criterion validity, this study evidences the existence of an indirect relationship between the communication score and job insecurity among nursing professionals during the COVID-19 pandemic. This finding is consistent with a study conducted in the United States during the early outbreaks of the pandemic. It found that effective and frequent organizational communication was crucial in mitigating job insecurity and improving employee well-being in times of crisis [18]. Another study from Belgium found a negative association between organizational communication and job insecurity [17].

This could mean that clear communication with employees concerning their expectations is key in the development of organizational commitment and the employee–organization relationship [19]. Thus, organizational measures, such as establishing efficient communication channels, fostering a supportive climate, and adjusting the organizational practices and policies, indirectly mitigate the stressful nature of job insecurity among employees [35], strengthening their commitment to the organization.

Unexpectedly, as we did not find previous reports in the literature, the score for the empathy subscale was a predictor of job insecurity. A couple of reasons behind this could be that empathy is one of the factors that mitigate exhaustion in health staff and is a unique psychological means having an impact not only on them but also on patients [36]. Alignment with institutional objectives allows employees to better understand the aspects involved in achieving them, leading them to strengthen their work commitment and improve their job performance [37].

One finding across all the regression models performed was the indirect association of job satisfaction with job insecurity. This association was reported in a bibliometric review [38] and confirmed by a recent multicenter study conducted in the United States, South Africa, and Croatia [39]. Nevertheless, a study carried out with workers of a mining company in Indonesia pointed out that job insecurity had a positive effect on job satisfaction [40]. The authors of the study explain that, under certain conditions, job insecurity could motivate workers to see the positive side of their work as a way of adapting to uncertainty.

In contrast, another study highlighted that stress, and not the working conditions and job insecurity, impacts nursing professionals’ job satisfaction [41]. In addition, another finding of the present study was that unstable contracts are connected with job insecurity, which confirms previous results showing that temporary contracts weaken the quality of life and increase job insecurity among nurses [42]. This is linked to research establishing positive relationships between job and life satisfaction among professionals in the nursing field [43]. Therefore, although the evidence from this study supports the indirect relationship between the two variables, further research is needed to elucidate the connection of these variables with intervening variables, such as working conditions, mental health issues, and motivation, in other contexts. Furthermore, job satisfaction is a complex phenomenon with multiple predictors, and understanding it is crucial for improving retention and well-being in nursing [44].

From a theoretical perspective, this study is a significant advance in the understanding of the CSS in the specific Peruvian cultural context. Particularly, this contribution is valuable considering the lack of validated scales measuring this construct in Peru. Furthermore, exploring the relationship of this scale with other variables in future studies may offer an opportunity to identify unique aspects of communication in the nursing profession. Cultural, socioeconomic, and health system factors may play a pivotal role in these investigations. This scale can also facilitate the assessment of communication skills training programs, allowing for a more accurate alignment between theory and practice in the Peruvian environment.

The practical implications of this study are substantial for human resource management in nursing. The proven association of job dissatisfaction with factors such as communication, empathy, contract type, and job satisfaction encourage multidimensional interventions. This study underlines the importance of healthcare institutions implementing training programs in communication skills and empathy, while also offering more stable contracts and improving working conditions with recognition programs and professional development opportunities. In this way, healthcare staff will experience a better quality of life, which will positively impact institutional indicators, enhance the quality of patient care, and improve job satisfaction [6].

This study represents one of the first psychometric explorations of the CSS in the Peruvian context. While the current sample provides fundamental data for the study, it is recommended that future research expand both the size and diversity of the sample, including nursing professionals from other regions of Peru, various levels of healthcare, and different specialties, to enhance the generalizability and transferability of the results. Considering that female participants constituted 93.3% of the sample, future studies should aim for a more balanced gender proportion to explore how gender influences the relationship between communication skills and job insecurity. Although self-report questionnaires are an effective means of gathering information, it is suggested that future research incorporate observational methods, such as role-playing or simulated patient interactions, to more accurately assess communication skills.

When analyzing the relationship between communication skills and job insecurity, a deeper exploration of mediating variables, such as mental health status, work environment, and team support, is recommended, as this could provide a more comprehensive understanding of these relationships. Given the significant impact of pandemics on the healthcare system, conducting long-term follow-up studies to observe the prolonged effects of communication skills training on nurses’ job security and other work-related outcomes is advisable. Additionally, it would be valuable to longitudinally explore how changes in communication skills and job insecurity over time affect the well-being and job satisfaction of nursing professionals, especially during periods of crisis and organizational stress. Finally, the development of interventions aimed at enhancing nurses’ communication skills and working conditions could be a fruitful area for future research and initiatives in health service management.

We identified four limitations in this study that are worth noting. First, since we relied solely on the existing theoretical framework for the factor structure, we may have overlooked specific cultural or contextual aspects that could affect the structure of the CSS in the study population. While this theoretical model is widely used across various contexts, we implemented a thorough process of cultural adaptation to mitigate this bias. Second, given that we used a non-probability sampling method, caution is advised when applying the results of this study to different demographic or cultural groups. Third, we did not perform a factor invariance analysis due to a considerable imbalance in the gender dimension, with women representing 93.3% of the participants. This limits our ability to determine whether the reported factor structure is consistent across different gender subgroups. Lastly, the self-reporting of the CSS without direct observation of behaviors may have limited the accuracy of the measurements, as the skills assessed are complex and often manifest in face-to-face interactions. Additionally, being based solely on self-reports through an online form due to COVID-19 pandemic restrictions, the study may be subject to biases related to self-perception and social desirability, which could affect the results. Nevertheless, during the data collection process, participants were informed about the importance of the study and assured of the confidentiality of their responses.

## 5. Conclusions

This study analyzed the psychometric properties of the CSS among Peruvian nurses. After a process of cultural adaptation and validation, we concluded that this four-factor scale maintains a structure similar to the original scale, demonstrating transcultural stability. The robust psychometric properties and adequate reliability support its applicability among Peruvian nurses. Furthermore, we confirmed that the subscales of the CSS (empathy, respect, communication skills, and social skills) can be used independently, providing flexibility in their application according to specific assessment needs.

The regression analysis revealed significant findings regarding job insecurity during the COVID-19 pandemic. Specifically, the type of contract emerged as a significant predictor of job insecurity, while job satisfaction showed an inverse relationship with it. Additionally, communication skills and empathy play a crucial role in the perception of job insecurity. These results confirm the criterion validity of the CSS through its association with job dissatisfaction.

The implications of this study are substantial for nursing practice. Further validations of this scale are recommended, considering nursing specialties or types of healthcare facilities. The results also highlight the need to implement communication skills development programs in nursing education and practice. Healthcare institutions should consider these findings to develop policies that improve working conditions and reduce job insecurity. Finally, the validated scale can serve as a valuable tool for assessing and improving communication competencies among Peruvian nursing staff, particularly during health crises.

## Figures and Tables

**Figure 1 healthcare-12-02582-f001:**
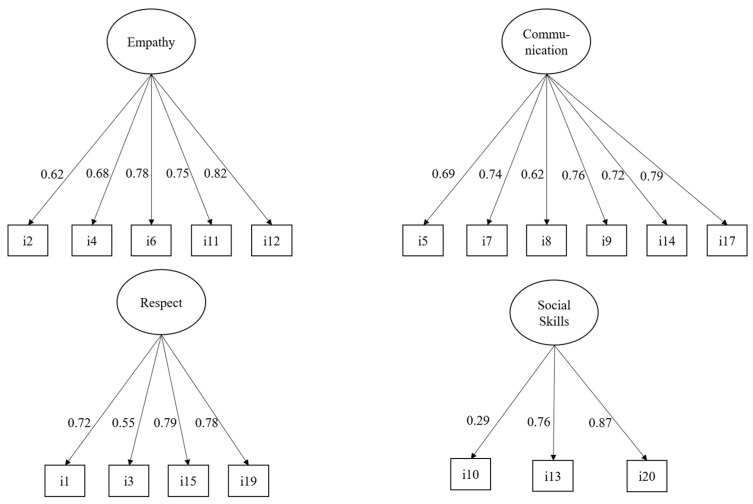
Confirmatory factor models for each subscale of the CSS.

**Figure 2 healthcare-12-02582-f002:**
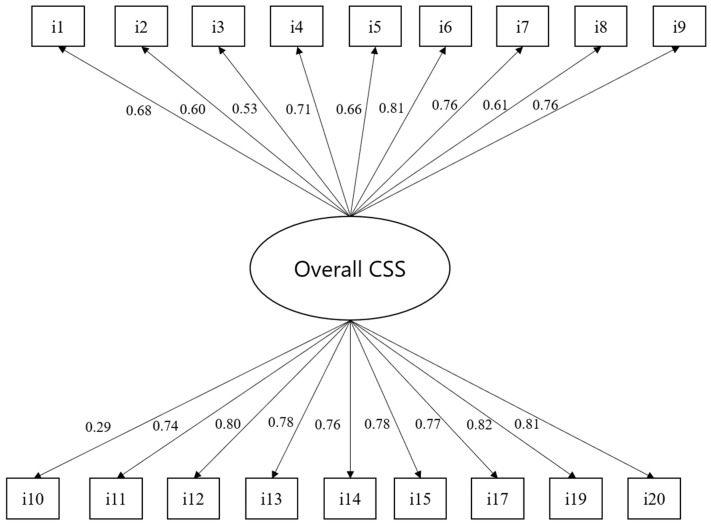
Confirmatory factor model for the entire CSS.

**Table 1 healthcare-12-02582-t001:** Means, standard deviations, minimums, and maximums of the CSS.

	M	SD	Min	Max
1. I respect my patient’s right to express themselves freely	4.67	0.97	2	6
2. I explore my patients’ emotions.	4.38	0.88	2	6
3. I respect the autonomy and freedom of the patients (patients’ decision-making).	4.34	0.99	1	6
4. When my patient talks to me, I show interest through body language (nodding, eye contact, smiling, touching the shoulder or hand).	4.53	1.02	1	6
5. I provide information to patients/family members (when my professional competency allows) about what concerns them.	4.42	1.01	1	6
6. I listen to patients without prejudice, regardless of their physical appearance, manners, or form of expression.	4.58	0.96	1	6
7. I clearly express my opinions to patients, considering their beliefs.	4.38	0.97	1	6
8. When I give information, I use pauses so that the patient can better understand what I am saying.	4.27	1.02	1	6
9. When I provide information to my patients, I do so in simple terms and with examples.	4.49	0.87	1	6
10. When a patient does something I disagree with (because it affects their health), I express my disagreement or concern.	3.87	1.12	1	6
11. I take time to listen to and understand my patients’ needs.	4.40	0.84	2	6
12. I try to understand my patients’ feelings and emotions.	4.47	0.83	2	6
13. When interacting with my patients, I listen to their feedback to communicate assertively.	4.46	0.80	1	6
14. I believe that the patient has the right to receive health information.	4.61	0.88	2	6
15. I feel that I respect patients’ needs, such as privacy.	4.58	0.82	2	6
16. It is difficult for me to be demanding or strict with patients.	3.87	1.09	1	6
17. I make sure that my patients have understood the information provided.	4.42	0.79	2	6
18. I find it hard to ask patients for information.	4.75	1.10	1	6
19. I listen to patients without prejudice, regardless of my beliefs.	4.51	0.82	2	6
20. I provide information in a clear and calm way so that my patient can understand the message I am conveying.	4.47	0.76	3	6

Note: Items were directly translated from Spanish to English.

**Table 2 healthcare-12-02582-t002:** Regression model predicting the levels of job insecurity based on the levels of the CSS.

	b	SE	B
Intercept	3.88 ***	0.32	
Contract type (services)	0.84 ***	0.11	0.53
Contract type (tenured)	0.09	0.18	0.03
Contract type (fixed-term)	0.56 **	0.19	0.17
Contract type (fees)	0.84 ***	0.18	0.28
Overall CSS	−0.01 **	0	−0.17
Job satisfaction (neither satisfied nor dissatisfied)	−0.02	0.1	−0.01
Job satisfaction (dissatisfied)	0.40 *	0.19	0.12
R^2^	0.25 ***		

Note: The reference category for contract type is “tenured-employee contract”, and the reference category for job satisfaction is “satisfied”. * *p* < 0.05, ** *p* < 0.01, and *** *p* < 0.001. b: unstandardized coefficient; SE: standard error; *p*: significance.

**Table 3 healthcare-12-02582-t003:** Regression model predicting the levels of job insecurity based on the subscales of the CSS.

	b	EE	B
Intercept	3.88 ***	0.29	
Contract type (services)	0.84 ***	0.11	0.53
Contract type (tenured)	0.10	0.18	0.03
Contract type (fixed-term)	0.54 **	0.19	0.16
Contract type (fees)	0.84 ***	0.18	0.28
Empathy	−0.04 **	0.01	−0.18
Job satisfaction (neither satisfied nor dissatisfied)	−0.02	0.1	−0.01
Job satisfaction (dissatisfied)	0.39 *	0.19	0.11
R^2^	0.26 ***		

Note: The reference category for contract type is “tenured-employee contract”, and the reference category for job satisfaction is “satisfied”. * *p* < 0.05, ** *p* < 0.01, and *** *p* < 0.001. b: unstandardized coefficient; SE: standard error; *p*: significance.

## Data Availability

Data are available upon request from the corresponding author.

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
