# Peer review of "Psychometric Assessment of the Communication Skills Scale Among Peruvian Nurses and Factors Associated with Job Insecurity"

_healthcare, 2024, doi:10.3390/healthcare12242582_

Round 1
Reviewer 1 Report
Comments and Suggestions for Authors
The aim of the AUTHORS was to evaluate the psychometric properties of the Communication Skills Scale (CSS) among Peruvian nurses and its association with job insecurity during the COVID-19 pandemic.
THEY: (1) explored five models of confirmatory factor analysis for the CSS and its four subscales and assessed the convergent validity and criterion validity of the scale by analyzing its connection with job insecurity through stepwise multiple regression. (2) Used insights from a focus group for the cultural adaptation of the scale. In the psychometric phase, 225 nurses participated through a virtual survey.
THEIR psychometric analysis revealed that the CSS and its subscales have a robust internal structure—similar to the original questionnaire—and are optimally reliable in the Peruvian population. Furthermore, the results show that job insecurity was associated with contract type, communication skills, empathy, and job satisfaction.
THEY concluded that the CSS and its subscales are valid and reliable to be applied to Peruvian nurses. THEY also suggest that Initiatives should be undertaken to strengthen communication skills and increase job satisfaction among nursing professionals by improving their working conditions, especially in times of crisis, to reduce job insecurity and promote well-being.
The ms is interesting and is in the field of the journal.
I have the following comments for the authros:
· The introduction is well set, however it could be slightly enriched by reporting some studies on the negative effect of the infodemic experienced during the pandemic. This would help to further enhance the study itself and the necessity of the proposed approach.
· In addition to this, in the introduction it is necessary to clearly develop both the key questions and the purpose (currently only nuanced) to give a clear message of your proposal.
· The study design presented in the methods should be presented a little better. This concerns not only the editorial structure that in some points should be improved in the exposition and with specific tools but also the content. It is suggested to first present the methodology with a small summary and then develop the themes using the standard MDPI headings. It is also suggested to pay attention to the content and not to insert introductory material in the methods (see lines 98-102; 114;140;178)
· In the results, table 1 must be enriched with the description of the items.
· Figures without a detailed description in the body of the manuscript are useless
· Considering the novelty of the study, it would be important to insert a paragraph in the discussion with suggestions for future developments
· The conclusions should be expanded to enhance the study, also using bullet points in one part with what emerges in pills.
Author Response
Dear Reviewer, we sincerely appreciate your comments and suggestions, which have greatly helped us improve our manuscript, especially in methods and results sections. Below, we present our responses to each of your observations:
Comment 1: The introduction is well set, however it could be slightly enriched by reporting some studies on the negative effect of the infodemic experienced during the pandemic. This would help to further enhance the study itself and the necessity of the proposed approach.
Response 1: We appreciate your suggestion and comment. Accordingly, we have added a paragraph to the introduction section. This information can be found on lines 45 to 51.
Comment 2: In addition to this, in the introduction it is necessary to clearly develop both the key questions and the purpose (currently only nuanced) to give a clear message of your proposal.
Response 2: Thank you for your insightful observation. We agree with your suggestion and have changed the last paragraph of the introduction according to your recommendations. This information can be found on lines 93 to 105.
Comment 3: The study design presented in the methods should be presented a little better. This concerns not only the editorial structure that in some points should be improved in the exposition and with specific tools but also the content. It is suggested to first present the methodology with a small summary and then develop the themes using the standard MDPI headings. It is also suggested to pay attention to the content and not to insert introductory material in the methods (see lines 98-102; 114;140;178).
Response 3: In response to your relevant suggestion, we have modified the methods section of the manuscript considering the journal’s instructions and your contributions.
Comment 4: In the results, table 1 must be enriched with the description of the items. Figures without a detailed description in the body of the manuscript are useless
Response 4: We appreciate your suggestion and have included a brief description of the most relevant aspects of each subscale; however, we refrained from going into too much detail in order to reduce the overall word count of the manuscript. Additionally, Figure 1 has been referenced in the text where it is described.
Comment 5: Considering the novelty of the study, it would be important to insert a paragraph in the discussion with suggestions for future developments
Response 5: We appreciate your suggestion and comment. Accordingly, we have added a paragraph to the discussion section. This information can be found on lines 401 to 423.
Comment 6: The conclusions should be expanded to enhance the study, also using bullet points in one part with what emerges in pills.
Response 6: We would like to reiterate our gratitude for these pertinent comments. In accordance with your suggestion, we have expanded the conclusions section of the manuscript. This information can be found on lines 442 to 463.
We once again thank you for your valuable suggestions and are confident that these improvements will significantly strengthen our work. We remain attentive to any additional comments you may have.
Sincerely,

Reviewer 2 Report
Comments and Suggestions for Authors
This is a paper discussing the relationship between communication skills of nursing professionals and their sense of job security. The author has systematically designed the article, proposing steps for cultural adaptation, psychometric properties, and correlational studies. I believe that the first step of adopting a cultural adaptation perspective is quite innovative. The paper demonstrates a deep understanding of the cultural adaptation process by collaborating with focus groups to adjust the scale. This not only enhances the content validity and conceptual equivalence of the scale but also increases the acceptance of the tool among the target population. Regarding psychometric properties, the paper provides a detailed confirmatory factor analysis of the CSS scale and its subscales, demonstrating good internal structure and reliability. These results help ensure the scientific rigor and broad applicability of the scale. In terms of correlational studies, the article explores the correlation between communication skills and job insecurity through multiple regression analysis, which is particularly relevant in the context of the pandemic and can provide empirical support for management and training.
However, the article may have some shortcomings that could be addressed for improvement.
First, expanding sample diversity. While the current sample provides foundational data for the study, it is recommended that future research increases both the sample size and diversity, including nurses from different regions and healthcare settings to enhance the generalizability and transferability of the results.
Second, the gender ratio imbalance. With female participants constituting as high as 93.3%, future studies should consider a more balanced gender ratio to explore how gender influences the relationship between communication skills and job insecurity.
Third, increasing observational studies. Although self-reported questionnaires are an effective way to gather information, it is suggested that future research incorporates observational methods, such as role-playing or simulated patient interactions, to more accurately assess communication skills.
Fourth, exploring mediating variables in depth. When analysing the relationship between communication skills and job insecurity, further exploration of mediating variables such as mental health status, work environment, and team support could provide a more comprehensive understanding of these relationships.
Fifth, conducting longitudinal studies. Given the ongoing development of the COVID-19 pandemic and its impact on the healthcare system, it is advisable to conduct long-term follow-up studies to observe the long-term effects of communication skills training on nurses' job security and other work-related outcomes.
These five points reflect my in-depth thoughts after reading the entire paper. If the author is unable to obtain data regarding the sample, I hope the above considerations can be incorporated into the final paragraph of the discussion section, supplementing the discussion with content that includes these five points.
Overall, this paper demonstrates in-depth research and practical recommendations at both academic and practical levels. With further improvements and expanded research, it will contribute to better serving the healthcare industry, especially during such a critical public health crisis.
Author Response
Dear Reviewer, we sincerely appreciate your comments and suggestions, which have greatly helped us improve our manuscript, especially in the discussion section. Below, we present our responses to each of your observations:
Comments:
First, expanding sample diversity. While the current sample provides foundational data for the study, it is recommended that future research increases both the sample size and diversity, including nurses from different regions and healthcare settings to enhance the generalizability and transferability of the results.
Second, the gender ratio imbalance. With female participants constituting as high as 93.3%, future studies should consider a more balanced gender ratio to explore how gender influences the relationship between communication skills and job insecurity.
Third, increasing observational studies. Although self-reported questionnaires are an effective way to gather information, it is suggested that future research incorporates observational methods, such as role-playing or simulated patient interactions, to more accurately assess communication skills.
Fourth, exploring mediating variables in depth. When analysing the relationship between communication skills and job insecurity, further exploration of mediating variables such as mental health status, work environment, and team support could provide a more comprehensive understanding of these relationships.
Fifth, conducting longitudinal studies. Given the ongoing development of the COVID-19 pandemic and its impact on the healthcare system, it is advisable to conduct long-term follow-up studies to observe the long-term effects of communication skills training on nurses' job security and other work-related outcomes.
These five points reflect my in-depth thoughts after reading the entire paper. If the author is unable to obtain data regarding the sample, I hope the above considerations can be incorporated into the final paragraph of the discussion section, supplementing the discussion with content that includes these five points.
Overall, this paper demonstrates in-depth research and practical recommendations at both academic and practical levels. With further improvements and expanded research, it will contribute to better serving the healthcare industry, especially during such a critical public health crisis.
Response: We appreciate your suggestion and comment. Accordingly, we have added two paragraphs to the discussion section. This information can be found on lines 401 to 423.
We once again thank you for your valuable suggestions and are confident that these improvements will significantly strengthen our work. We remain attentive to any additional comments you may have.
Sincerely,

Reviewer 3 Report
Comments and Suggestions for Authors
Thank you for the opportunity to provide a review of this article.
The paper has several shortcomings that must be addressed before being considered for publication.
The paper's objective is to validate the CSS scale in the Peruvian population, which is appropriate. However, the methodology is only correct up to a certain point. The data analysis, discussions, and conclusions need to be reconsidered.
1. You conducted a cultural adaptation and confirmatory analysis for validation, which is fine. But, for criterion validity, you performed a regression where the CSS scale predicts job insecurity. But how did you choose this scale “job insecurity” and why? I don’t think is appropriate because I can’t see how some communication skills can predict some feelings of fear regarding an insecure job. Is not theoretically relevant. Why did you choose this scale for this purpose? Where is the theoretical background? It doesn't seem to make sense.
2. If you want to assess criterion validity, I suggest using a different dependent variable that is more theoretically relevant. “Job satisfaction” might be a more suitable option. (I noticed you have in the article something about this). However, you’ll need to provide supporting arguments for this choice in the literature review.
3. Moreover, when you performing a regression model to determine if CSS can predict values of another scale, it is important to focus solely on these two scales. In Tables 2 and 3, you include in regression models several other variables. Why? Your purpose is to validate the scale, not to examine the impact of additional variables on the dependent variable. It is not the aim of the paper.
4. When you perform confirmatory analysis, if you want, you can use socio-demographic variables to check if the scale’s consistency holds across different population categories. To do this, you could perform a multigroup analysis or conduct separate confirmatory factor analyses for distinct groups.
5. Therefore, the discussion about correlations between various other variables does not align with the primary purpose of your article and appears irrelevant, unless you intend to have two separate objectives: one to validate the scale and another to demonstrate that values of the CSS impact other variables.
6. Based on these points you need to revise the abstract, data presentation, conclusions, and discussion sections.
7. In addition,
a. Lines 200-202. It is not clear why you excluded Items 16 and 18.
b.Lines 224. Item 10 needs to be deleted, the value of 0.29 is too low
c.Tables 2, and 3 need to be revised in line with the previous feedback. Moreover, an R2 value of 0.25 or 0.26 is low. If you delete the additional variables (other than CSS) these values probably decrease. (It is not a predictive relationship)
I wish you success with your article!
Author Response
Dear Reviewer, we sincerely appreciate your comments and suggestions, which have greatly helped us improve our manuscript, especially in methods and discussion sections. Below, we present our responses to each of your observations:
Reviewer:
Thank you for the opportunity to provide a review of this article.
The paper has several shortcomings that must be addressed before being considered for publication.
The paper's objective is to validate the CSS scale in the Peruvian population, which is appropriate. However, the methodology is only correct up to a certain point. The data analysis, discussions, and conclusions need to be reconsidered.
Comment 1: You conducted a cultural adaptation and confirmatory analysis for validation, which is fine. But, for criterion validity, you performed a regression where the CSS scale predicts job insecurity. But how did you choose this scale “job insecurity” and why? I don’t think is appropriate because I can’t see how some communication skills can predict some feelings of fear regarding an insecure job. Is not theoretically relevant. Why did you choose this scale for this purpose? Where is the theoretical background? It doesn't seem to make sense.
Response 1: Thank you for your insightful observation. In response, we have added a paragraph to the introduction that supports the relationship between communication and job insecurity, based on the Conservation of Resources (COR) theory proposed by Hobfoll. This information can be found on lines 80 to 92.
Comment 2: If you want to assess criterion validity, I suggest using a different dependent variable that is more theoretically relevant. “Job satisfaction” might be a more suitable option. (I noticed you have in the article something about this). However, you’ll need to provide supporting arguments for this choice in the literature review.
Response 2: This is an interesting comment; however, it is not feasible to consider job satisfaction as a variable for determining criterion validity for two reasons. First, job satisfaction was measured with a single general question that does not capture all its dimensions. Second, the measurement scale for job satisfaction is ordinal categorical, requiring the use of ordinal logistic regression, which complicates data interpretation. Given these limitations in both the measurement validity of the variable and its ability to yield interpretable results, we decided not to include it as a dependent variable.
Comment 3: Moreover, when you performing a regression model to determine if CSS can predict values of another scale, it is important to focus solely on these two scales. In Tables 2 and 3, you include in regression models several other variables. Why? Your purpose is to validate the scale, not to examine the impact of additional variables on the dependent variable. It is not the aim of the paper.
Response 3: After careful consideration of this comment, we maintain our original position. Evaluating the association between communication skills and job insecurity in a model that includes only this variable, for which a measurement is still being developed, could result in a spurious correlation. Therefore, we estimated a stepwise regression model to integrate variables that provide the greatest statistical control, enabling us to better understand the true association between communication and job insecurity. Although exploring the association between job insecurity and other variables is not the primary focus of our study, we have decided to retain this approach to strengthen the criterion validity of our research. Additionally, we are introducing an objective related to exploring the factors associated with job insecurity.
Comment 4: When you perform confirmatory analysis, if you want, you can use socio-demographic variables to check if the scale’s consistency holds across different population categories. To do this, you could perform a multigroup analysis or conduct separate confirmatory factor analyses for distinct groups.
Response 4: This is an interesting comment. However, multigroup analysis, which evaluates the invariance of the scale across different subpopulations in the study, requires a larger sample size for each subgroup than what is currently available in this study. Therefore, this limitation has been addressed in the research.
Comment 5: Therefore, the discussion about correlations between various other variables does not align with the primary purpose of your article and appears irrelevant, unless you intend to have two separate objectives: one to validate the scale and another to demonstrate that values of the CSS impact other variables.
Response 5: We agree with your suggestion, and for that reason, we have modified the aim of the study as follows: The aim of this study was to evaluate the psychometric properties of the Communication Skills Scale (CSS) among Peruvian nurses, as well as the factors associated with job insecurity during the COVID-19 pandemic. Consequently, the title of the manuscript has also been modified.
Comment 6: Based on these points you need to revise the abstract, data presentation, conclusions, and discussion sections.
Response 6: We agree with your suggestion. Consequently, all sections of the manuscript have been reviewed and modified.
Comment 7: In addition.
- Lines 200-202. It is not clear why you excluded Items 16 and 18.
- Lines 224. Item 10 needs to be deleted, the value of 0.29 is too low
Response 7: In response to your suggestion, we have provided a clearer explanation of why items 16 and 18 were removed, as well as the rationale for retaining item 10.
Comment 8: In addition.
- Tables 2, and 3 need to be revised in line with the previous feedback. Moreover, an R2 value of 0.25 or 0.26 is low. If you delete the additional variables (other than CSS) these values probably decrease. (It is not a predictive relationship)
Response 8: We appreciate the reviewer's observation regarding the R² value. However, we would like to offer a different perspective supported by the following evidence: An R² of 0.25 represents a variance explanation of 25%, which, while perhaps modest in basic sciences, holds substantial significance in health and social sciences research. According to Cohen's established criteria, this magnitude constitutes a large effect size in our field. This interpretation is particularly relevant given the inherent measurement challenges and error typically associated with survey-based psychometric research.
We once again thank you for your valuable suggestions and are confident that these improvements will significantly strengthen our work. We remain attentive to any additional comments you may have.
Sincerely,
